# An integrative approach to prioritize candidate causal genes for complex traits in cattle

**Mohammad Ghoreishifar**[1,2]*, **Iona M. Macleod**[1,2], **Amanda J. Chamberlain**[1,2], **Zhiqian Liu**[1], **Thomas J. Lopdell**[3,4], **Mathew D. Littlejohn**[3,4], **Ruidong Xiang**[1,5], **Jennie E. Pryce**[1,2], **Michael E. Goddard**[1,5]

1 Agriculture Victoria Research, AgriBio Centre for AgriBioscience, Bundoora, Victoria, Australia, 2 School of Applied Systems Biology, La Trobe University, Bundoora, Victoria, Australia, 3 Research and Development, Livestock Improvement Corporation, Hamilton, New Zealand, 4 AL Rae Centre for Genetics and Breeding, Massey University, Hamilton, New Zealand, 5 Faculty of Veterinary & Agricultural Science, University of Melbourne, Parkville, Victoria, Australia

* mohammad.ghoreishifar@agriculture.vic.gov.au

## Abstract

Genome-wide association studies (GWAS) have identified many quantitative trait loci (QTL) associated with complex traits, predominantly in non-coding regions, posing challenges in pinpointing the causal variants and their target genes. Three types of evidence can help identify the gene through which QTL acts: (1) proximity to the most significant GWAS variant, (2) correlation of gene expression with the trait, and (3) the gene's physiological role in the trait. However, there is still uncertainty about the success of these methods in identifying the correct genes. Here, we test the ability of these methods in a comparatively simple series of traits associated with the concentration of polar lipids in milk. We conducted single-trait GWAS for ~14 million imputed variants and 56 individual milk polar lipid (PL) phenotypes in 336 cows. A multi-trait meta-analysis of GWAS identified 10,063 significant SNPs at FDR ≤ 10% ($P \le 7.15E$-5). Transcriptome data from blood (~12.5K genes, 143 cows) and mammary tissue (~12.2K genes, 169 cows) were analyzed using the genetic score omics regression (GSOR) method. This method links observed gene expression to genetically predicted phenotypes and was used to find associations between gene expression and 56 PL phenotypes. GSOR identified 2,186 genes in blood and 1,404 in mammary tissue associated with at least one PL phenotype (FDR ≤ 1%). We partitioned the genome into non-overlapping windows of 100 Kb to test for overlap between GSOR-identified genes and GWAS signals. We found a significant overlap between these two datasets, indicating that GSOR-significant genes were more likely to be located within 100 Kb windows that include GWAS signals than those that do not ($P = 0.01$; odds ratio = 1.47). These windows included 70 significant genes expressed in mammary tissue and 95 in blood. Compared to all expressed genes in each tissue, these genes were enriched for lipid metabolism gene ontology (GO). That is, seven of the 70 significant mammary transcriptome genes ($P < 0.01$; odds ratio = 3.98) and

**Data availability statement:** Data generated or analyzed during this study are included in this published article and its supplementary files. All gene expression data was taken from previously published studies as detailed in the materials and methods. The Polar Lipids data is publicly available in the Dryad repository. https://doi.org/10.5061/dryad.bcc2fqzph.

**Funding:** This study was undertaken as part of the DairyBio program, which is jointly funded by Dairy Australia (Melbourne, Australia), Agriculture Victoria (Melbourne, Australia), and The Gardiner Foundation (Melbourne, Australia). The funders had no role in study design, data collection and analysis, decision to publish, or preparation of the manuscript.

**Competing interests:** The authors have declared that no competing interests exist.

five of the 95 significant blood genes ($P < 0.10$; odds ratio = 2.24) were involved in lipid metabolism GO. The candidate causal genes include *DGAT1*, *ACSM5*, *SERINC5*, *ABHD3*, *CYP2U1*, *PIGL*, *ARV1*, *SMPD5*, and *NPC2*, with some overlap between the two tissues. The overlap between GWAS, GSOR, and GO analyses suggests that together, these methods are more likely to identify genes mediating QTL, though their power remains limited, as reflected by modest odds ratios. Larger sample sizes would enhance the power of these analyses, but issues like linkage disequilibrium would remain.

## Author summary

Complex traits are influenced by many genetic variants. Although some of these variants have been mapped to a genomic region, it has proven difficult to identify the causal mutation or the gene that mediates its effect on the trait. We combined three sources of evidence to identify genes causing variation in the concentration of polar lipids (PLs) in cows' milk. We chose these phenotypes because we assumed that genes affecting these traits would have known physiological functions. Indeed, we aimed to identify the associated genes and provide further evidence to support that these genes are the likely relevant genes. We examined the association between the concentration of PLs and > 14,000,000 genome-wide genetic variants, of which about 0.07% were significantly associated with the PL phenotypes. We also examined the association between the PL phenotypes and the activity of > 13,000 genes, of which around 10% were significantly associated with the PL phenotypes. Genome-wide analyses showed that the identified genes and genetic variants occurred near each other in the genome. The conclusion that these genes cause variation in PLs concentration was supported by the finding that a higher proportion of these genes, compared to other genes, are involved in lipid metabolism.

## Introduction

Genome-wide association studies (GWASs) test the statistical association between millions of variants, such as single nucleotide polymorphisms (SNP), and a complex trait. While GWAS have been effective in identifying numerous quantitative trait loci (QTL), it is difficult to be certain which are the specific causal variants and target genes through which they influence phenotypes [1–4], mainly because of the small size of most effects, the high degree of linkage disequilibrium (LD) among nearby variants, and the fact that the majority of QTL are located in non-coding regions of the genome. Three types of evidence can be used to identify the gene through which QTL work: (1) genes near the most significant GWAS variant, (2) genes whose expression is correlated with the trait, and (3) genes whose physiological role is related to the trait [5]. Although none of these three types of evidence is conclusive, if

they all point to the same genes, that would be good evidence that the identified genes were correct. This paper examines the extent to which these three types of evidence agree and how often the nearest gene to QTL is the likely candidate causal gene.

Non-coding GWAS loci are likely to influence quantitative traits by modulating the expression of their target genes, which are referred to as expression QTL (eQTL). There are two types of eQTL: (1) *Cis*-eQTLs are regulatory variants that influence the expression of their target genes located not only on the same chromosome but also within a nearby genomic region, typically within 1 Mb of a gene's transcription start site (TSS); (2) *trans*-eQTLs regulate the expression of target genes located on different chromosomes or far from the variant on the same chromosome.

As mentioned earlier, genes whose expression correlates with a trait may be genes through which QTL affects that trait. One approach to identifying these genes is the transcriptome-wide association study (TWAS). TWAS combines an expression reference panel (individuals with both gene expression and genotype data) with a GWAS dataset (individuals with phenotype and genotype data) to uncover gene-trait associations [1,2]. In fact, TWAS uses the expression reference panel to train per-gene predictive expression models using *cis* variants, i.e., SNPs typically located within 500 kb to 1 Mb of the gene's TSS [1,2]. Predicted gene expression can then be calculated for GWAS cohorts with available genotype and phenotype data [1,2], and the correlation between phenotype and predicted gene expression is calculated, which is commonly referred to as TWAS [1–4].

Another way to identify genes whose expression is (genetically) related to a phenotype is to estimate genetic correlations between gene expression and the trait [6]. A recently introduced method that estimates the correlation between observed gene expression and genetically predicted phenotypes is referred to as genetic score omics regression (GSOR) [7]. This method uses genomic breeding values (calculated from a prediction equation based on a panel of genomic variants) and examines their association with gene expression levels in a given tissue. This linear combination of all of the SNP effects is referred to as genomic estimated breeding values (GEBVs) in animal genetics or polygenic scores (PGSs) in human genetics [8]. Usually, the most significant effects on gene expression are *cis* effects, so to exploit this fact, GSOR uses the local component of this GEBV/PGS (i.e., accumulated effect of SNPs typically located within 1 Mb of the gene's TSS) and then calculates the correlation between this local GEBV and expression of the gene. Therefore, an advantage of GSOR over TWAS is that the local GEBV/PGS is used as the response variable.

Functional knowledge of the role of the gene is the third type of evidence for the gene mediating the QTL. Functional knowledge of genes is recorded in databases such as the Gene Ontology (GO) database [9,10]. We expect the list of genes affecting a particular trait will be enriched for functional annotations relevant to that trait. This enrichment should be clearer for physiologically simpler traits (e.g., human bone mineral density [5] or milk composition in cattle) than traits such as milk yield, which are influenced by many physiological pathways. Here, we have chosen the concentration of various polar lipids (PL) in milk as our phenotypes, with the hypothesis that these traits are expected to have simpler genetic architecture than many other complex phenotypes.

Milk polar lipids, comprising mainly phospholipids, sphingolipids, and glycosphingolipids, represent <2% of total fat and are located primarily in the fat globule membrane [11]. Phospholipids phosphatidylcholine (PC), phosphatidylethanolamine (PE), phosphatidylserine (PS), and phosphatidylinositol (PI) and sphingolipids sphingomyelin (SM) are the major classes of polar lipids present in milk, whereas glycosphingolipids such as lactosylceramide (LacCer) and glucosylceramide (GluCer) are found in much lower concentration [11].

All three of these types of evidence are imperfect in that they are still subject to false positive or false negative findings. The extent of the overlap between the genes identified by each method provides evidence of the power of all methods. Also the genes identified by all methods are the most likely correct [5].

The specific objectives of this study were: (1) to identify genomic regions associated with PL phenotypes based on a meta-analysis of multi-trait GWAS (i.e., GWAS$_{Meta}$); (2) to identify genes from white blood cells (WBC) and mammary tissues, whose expression are significantly associated with the PL phenotypes as inferred from GSOR; (3) to perform

gene list enrichment analysis of all significant GSOR genes (hereinafter referred to as GSOR hits) to identify gene ontology (GO) terms potentially involved in the regulation of milk PL, and (4) to investigate the overlap between GWAS and GSOR; and (5) to investigate enrichment of GSOR hits proximal to GWAS$_{Meta}$ signals in relevant GO terms. These sets of analyses can provide evidence of causality from three different sources of information: GWAS, gene expression, and the physiological role of those genes [5].

## Results

### Heritability estimation and GWAS for PL phenotypes

The heritability (±SE) estimated for 56 PL phenotypes using REML ranged from 0.07 ± 0.12 to 0.7 ± 0.13. We presented these results in S1 Data. Single-trait GWAS was performed for 56 PL phenotypes using 336 animals and 14,056,074 imputed variants (Table 1). A total of 9,923 variants demonstrated significant associations across 20 individual PL phenotypes (P ≤ 2.78E-5; FDR ≤ 10%), while no significant associations were identified for the remaining PL. Among the significant associations, ~78% were located on just two chromosomes, including BTA 24 (41%) and BTA 14 (37%). Genomic inflation (λ) of the GWAS test statistics ranged from 0.94 to 1.07, with an average of 1.01 across the 56 PLs (S1 Data).

Summary statistics from GWAS$_{Meta}$ (see Materials and Methods) for 56 PL phenotypes are provided in S2 Data and the Manhattan plot in Fig 1. In the GWAS$_{Meta}$, 10,063 SNPs were significant (P ≤ 7.15E-5; FDR ≤ 10%), of which 13% were

**Table 1. Data description of this study.**

| Data | Analysis | N samples Genotyped | N Variants | Phenotype | EBV | Gene Expression |
|---|---|---|---|---|---|---|
| GWAS dataset | MLM LOCO GWAS (GCTA [35]) | 336 | 14,056,074 | 56 milk PL | – | – |
| MLM LOCO GWAS results | Multi-trait GWAS [18] | 336 | 14,056,074 | 56 milk PL | – | – |
| GWAS dataset | Prediction equations (BayesR3 [50]) | 336 | 1,236,780 | 56 milk PL | – | – |
| RNA-seq WBC | Transcriptome Wide Association Study (GSOR [7]) | 143 | 1,236,780 | – | Predicted local EBV | 12,533 gene expression |
| RNA-seq Mammary | Transcriptome Wide Association Study (GSOR [7]) | 169 | 1,236,780 | – | Predicted local EBV | 12,237 gene expression |

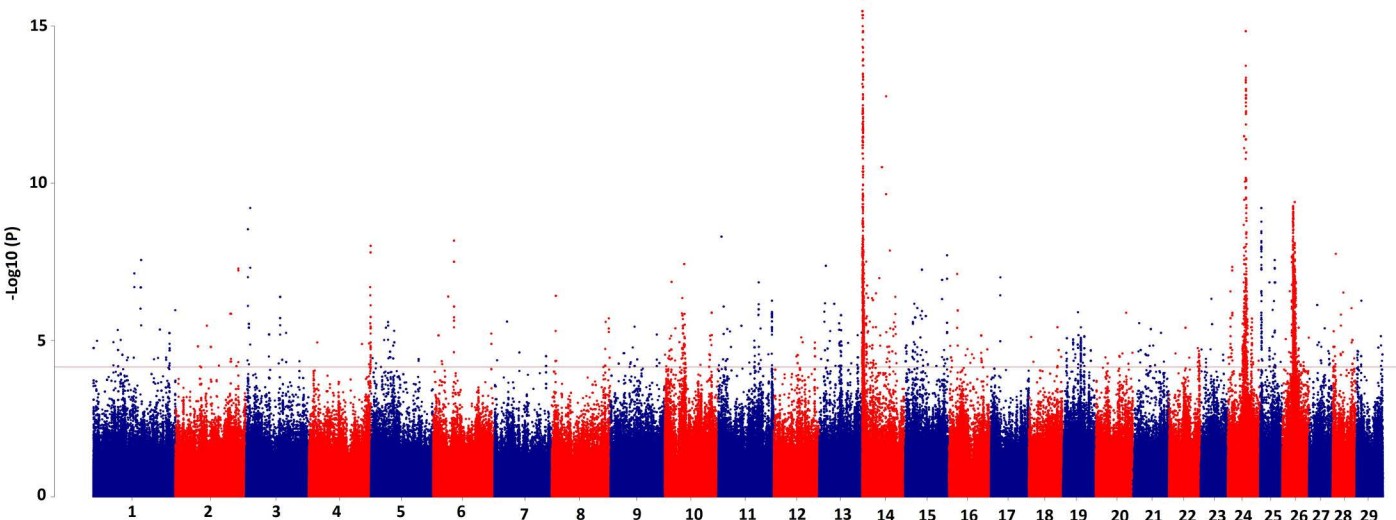

**Fig 1. Manhattan plot for multi-trait meta-analyses GWAS for ~14 million imputed variants and 56 species of polar lipids in milk.**

positioned on BTA 24, 27% on BTA 14, less than 1% on BTA 29 and 21% on BTA 26. This difference between $GWAS_{Meta}$ and single-trait GWAS regarding the distribution of significant SNPs in various autosomes is likely because significant associations were found for only 20 individual PL phenotypes, whereas all 56 PL phenotypes were included in the $GWAS_{Meta}$ analyses.

### GSOR analysis

We tested the association between the expression of 12,237 genes in the mammary dataset and local GEBVs for 56 species of PL (*p*-values are presented in S3 Data). Collectively, we found 1,404 genes that were significantly associated with at least one PL phenotype (S4 Data). The number of mammary GSOR hits across individual PL phenotypes ranged from 136 to 212, with an average of 173 genes per trait.

We tested the association between the expression of 12,533 genes in the WBC dataset and local GEBVs for 56 species of PL (*p*-values are presented in S5 Data). In total, 2,186 genes were identified that were associated with at least one PL (S6 Data). The number of WBC GSOR hits across individual PL traits ranged from 277 to 349 genes, averaging 314 genes per trait.

### Functional annotation of gene lists

Using the DAVID database [9,10], we conducted gene list enrichment analyses for 1,404 mammary GSOR hits versus 12,237 background genes and for 2,186 WBC GSOR hits versus 12,533 background genes (S3, S4, S5, and S6 Data). Results are presented in Table 2. For the mammary RNA-seq data, significant enrichment was observed for the GO term lipid metabolism with 59 genes (FDR < 0.01) listed with this term and GPI-anchor biosynthesis with eight genes (FDR < 0.05). For the WBC RNA-seq data, significant enrichment was observed for the GO terms cell adhesion (36 genes; FDR < 0.05), immune response, positive regulation of T cell-mediated cytotoxicity, and cell-cell adhesion (FDR < 0.05).

### Do GSOR hits agree with $GWAS_{Meta}$ signals?

We investigated the agreement between GSOR hits and $GWAS_{Meta}$ signals using non-overlapping windows of 100 Kb and 500 Kb. We observed a significant overlap between GSOR hits and $GWAS_{Meta}$ signals with both tissues ($P \le 0.05$). These results are presented in Table 3. For example, using 100 Kb windows, we identified 24,869 non-overlapping windows, of which 839 included $GWAS_{Meta}$ signals. Of our 1,404 mammary GSOR hits, 70 were found within these 839 windows with $GWAS_{Meta}$ signals, resulting in a Fisher Exact Test *p*-value of 0.003 (odds ratio = 1.47). In addition, for the WBC data with the same window size, 95 out of the total number of 2,184 GSOR hits were positioned inside the 839 GWAS-Marked windows (See Material and Methods), resulting in a *p*-value of 0.024 (odds ratio = 1.28).

**Table 2. Gene list enrichment analyses of mammary and white blood cell GSOR genes.**

| Category | Term | N[1] | FDR | data |
|---|---|---|---|---|
| UP_KW_BIOLOGICAL_PROCESS | Lipid metabolism | 59 | 8.4E-3 | Mammary |
| UP_KW_BIOLOGICAL_PROCESS | GPI-anchor biosynthesis | 8 | 2.8E-2 | Mammary |
| UP_KW_BIOLOGICAL_PROCESS | Cell adhesion | 36 | 3.6E-2 | WBC |
| GOTERM_BP_DIRECT | Immune response | 52 | 7.5E-3 | WBC |
| GOTERM_BP_DIRECT | Positive regulation of T cell mediated cytotoxicity | 14 | 7.5E-3 | WBC |
| GOTERM_BP_DIRECT | Cell-cell adhesion | 34 | 1.1E-2 | WBC |

[1]Number of genes.

PLOS Genetics

**Table 3. Investigation of the agreement between GSOR hits and GWAS$_{Meta}$ signals using non-overlapping windows of various sizes.**

| RNA-seq data | Window size (Kb) | Total windows | GWAS-Marked windows[1] | GSOR hits in GWAS-Marked windows | Total GSOR hits | P-value[2] (odds ratio) |
|---|---|---|---|---|---|---|
| Mammary | 100 | 24,869 | 839 | 70 | 1,404 | 0.003 (1.47) |
| | 500 | 4,996 | 457 | 172 | 1,404 | 0.002 (1.33) |
| WBC | 100 | 24,869 | 839 | 95 | 2,184 | 0.024 (1.28) |
| | 500 | 4,996 | 457 | 241 | 2,184 | 0.025 (1.20) |

[1]non-overlapping windows that include (at least one) GWASMeta signal;

[2]Fisher exact Test p-value.

## Are GSOR hits located within GWAS-Marked windows enriched for the Gene Ontology term lipid metabolism?

We focused on a subset of GSOR hits located within GWAS-Marked windows and compared them with the background genes in terms of the proportion of genes involved in the lipid metabolism GO term found in each list. The results are presented in Table 4, with background genes listed in S3 & S5 Data. Mammary GSOR hits located in GWAS-Marked windows were enriched with lipid metabolism GO term with all window sizes ($P \leq 0.01$). This result for WBC data was close to the significance level ($P < 0.10$) using only 100 Kb windows. The best results (odds ratio) were obtained with 100 Kb non-overlapping windows. In mammary RNA-seq data, for example, 332 of 12,237 background genes contained the lipid metabolism GO term, while seven of the 70 GSOR hits contained this GO term. This results in a $P$-value of 0.003 with an odds ratio of 3.98. For the WBC data, 302 out of 12,533 background genes were listed with the lipid metabolism GO term, while five out of 95 GSOR hits within GWAS-Marked windows contained the same GO term ($P = 0.08$; odds ration = 2.24).

Therefore, GSOR hits from mammary and WBC data located in GWAS-Marked windows and known to be involved in lipid metabolism (based on GO annotation) were potential candidate causal genes as multiple sources of information supported their causality (Table 5). These candidate causal genes include *DGAT1, SERINC5, PIGL, CYP2U1, ABHD3, CSM5, and ARV1* from mammary gland and *DGAT1, PIGL, CYP2U1, SMPD5* and *NPC2* from WBC data. Fig 2 illustrates two examples (*DGAT1* and *SERINC5*), showing the mammary GSOR hits and background genes in their proximity, along with the Manhattan plot from GWAS$_{Meta}$.

**Table 4. Enrichment of GSOR hits located within GWAS-Marked windows in Lipid metabolism GO compared to background genes, using different non-overlapping window sizes.**

| RNA-seq data | Window size (Kb) | GSOR hits within GWAS-Marked windows[1] | | Background genes | | P-value[3] (odds ratio) |
|---|---|---|---|---|---|---|
| | | Involved in[2] | Not involved in[2] | Involved in[2] | Not involved in[2] | |
| Mammary | 100 | 7 | 63 | 332 | 11,905 | 0.003 (3.98) |
| | 500 | 11 | 161 | 332 | 11,905 | 0.008 (2.44) |
| WBC | 100 | 5 | 90 | 302 | 12,231 | 0.08 (2.24) |
| | 500 | 9 | 232 | 302 | 12,231 | 0.19 (1.57) |

[1]GWAS-Marked windows are non-overlapping windows of different length that contain at least one GWASMeta signal;

[2]Number of genes involved in and not involved in Lipid metabolism GO, respectively; 3 Fisher Exact Test p-value.

**Table 5. Candidate causal genes for PL traits that are a subset of GSOR genes located in GWAS-Marked windows of 100 Kb and involved in Lipid metabolism GO term.**

| Ensemble ID | Gene | Tissue | P-value (GSOR)[1] | BTA & position |
|---|---|---|---|---|
| ENSBTAG00000037449 | ACSM5 | M | 2.87E-5 | 25:18090665-18119714 |
| ENSBTAG00000001619 | SERINC5 | M | 7.97E-5 | 10:11111559-11216841 |
| ENSBTAG00000005709 | ABHD3 | M | 1.30E-5 | 24:34541295-34591097 |
| ENSBTAG00000012972 | CYP2U1 | M & WBC | 4.93E-5 | 6:17261763-17282966 |
| ENSBTAG00000013273 | PIGL | M & WBC | 2.65E-6 | 19:33284684-33326707 |
| ENSBTAG00000021822 | ARV1 | M | 2.66E-5 | 28:3714519-3737767 |
| ENSBTAG00000026356 | DGAT1 | M & WBC | 2.97E-5 | 14:603813-612791 |
| ENSBTAG00000015040 | SMPD5 | WBC | 3.86E-5 | 14:774643-776724 |
| ENSBTAG00000021955 | NPC2 | WBC | 1.14E-4 | 10:85774962-85783794 |

[1]The average GSOR p-value across the significantly associated PLs with the gene.

## Discussion

GWAS have identified many variants associated with complex traits, e.g., human height [12] or livestock production traits [13]. The causal variant may directly point to the associated gene if it is located in a coding region. However, studies have reported that most variants associated with complex traits by GWAS are located in non-coding regions [14,15] and therefore have unknown functions. QTL in non-coding regions can affect phenotype by regulating gene expression (i.e., eQTL) [16,17]. The challenge, mainly due to LD, is not only to pinpoint the actual causal variants in non-coding regions but also to find the target genes through which they affect phenotype [2,4,7]. Therefore, TWAS and other methods have been developed to prioritize causal genes at GWAS loci [1].

In this study, we integrated GWAS$_{Meta}$ with GSOR analyses to identify potential candidate causal genes affecting complex trait phenotypes (i.e., concentration of PLs in milk). Using meta-analyses on 56 single-trait GWAS (GWAS$_{Meta}$), 10,063 associations were identified. The GWAS$_{Meta}$ approach was used because meta-analyses can enhance the power to detect genetic variants by leveraging the shared genetic architecture across correlated traits [18]. Next, GSOR was performed to identify significant associations between gene expression and local GEBVs for PL phenotypes. Our GSOR analysis revealed that 2,186 and 1,404 genes from WBC and mammary transcriptome were significantly associated with at least one PL phenotype. However, not all gene-trait associations are causal. TWAS methods (including GSOR) often detect multiple significant genes per locus [2,5].

LD can contribute to TWAS false positives [2]. For instance, when an eQTL affecting phenotype is in LD with another eQTL, this will cause a correlation between the expression of both genes and the trait, but the relationship is not causal for the second gene [2]. In our study, both *DGAT1* and *SLC52A2* genes were GSOR hits, with the former being a well-known causal gene affecting milk fat [19–23] (Fig 2a). *SLC52A2* is probably a false GSOR hit because its expression is highly correlated (r = 0.5, p = 2.8E-12) with an expression of *DGAT1* in the mammary gland. Another example is the correlation of expression between *SERINC5* and *PSMC6* gene (r = -0.34; p = 3.7E-6); while the former was one of our prioritized candidate causal genes, the latter would be more likely a false hit (Fig 2b). Thus, assuming that gene expression mediates the genetic impact on complex traits, GSOR or TWAS associations do not offer direct evidence of causal links between gene expression and these traits. Instead, they represent associations between expression levels and phenotypes [2]. Therefore, additional sources of evidence are required to prioritize candidate causal gene-trait associations.

We assessed the agreement between GSOR hits and GWAS$_{Meta}$ signals. By partitioning the genome into non-overlapping windows of different sizes, including 0.1 and 0.5 Mb, we demonstrated that GSOR hits were significantly more likely to be located within GWAS-Marked windows. This significant overlap supports the hypothesis that GSOR hits include causal genes that mediate the effect of GWAS$_{Meta}$ signals on the PL phenotypes.

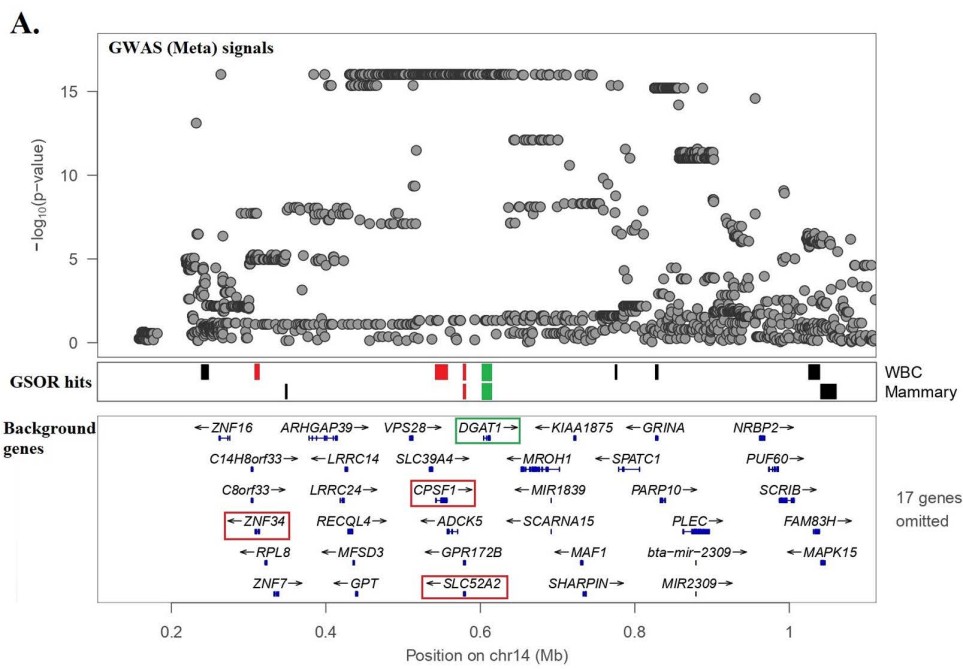

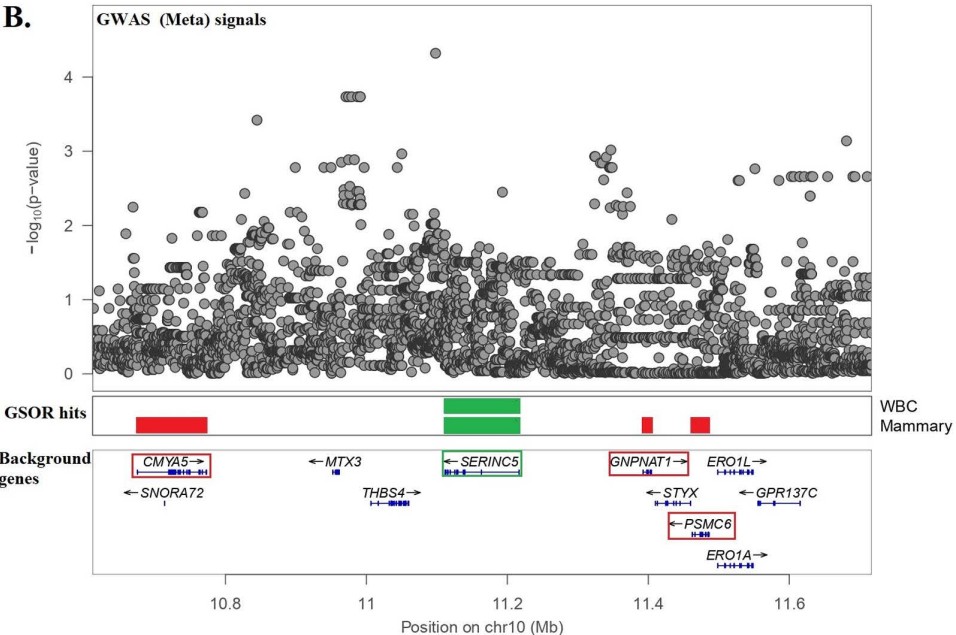

**Fig 2. GWAS and GSOR signals surrounding two of the prioritized genes, *DGAT1* and *SERINC5*.** The first level of the plot shows the Manhattan plot for GWAS meta-analysis, the second level shows the genes whose expression in mammary gland and/or WBC data were significantly associated with at least one PL phenotype (genes highlighted with green are potential candidate causal genes), and the third level plot represents background genes expressed in mammary data.

If GSOR hits are tagged by GWAS_{Meta} signals (i.e., those genes that are located within GWAS-Marked windows) and are truly causal, they should show enrichment for biologically relevant GO. Milk PLs are promising target phenotypes to test this hypothesis. Traits like milk yield are not simple, and many pathways likely influence the final phenotypes. However, identifying enriched pathways with physiologically simple phenotypes like milk PLs should be easier. Additionally, the medium to high heritability of most PL phenotypes increases the likelihood of detecting genetic associations, even with smaller sample sizes. We tested this hypothesis by investigating the enrichment of lipid metabolism GO terms for genes that were GSOR hits located within GWAS-Marked windows compared to total background genes. Results indicated a significant enrichment of these GSOR hits for lipid metabolism GO (Table 4), particularly in mammary tissue, where the 100 Kb windows showed the strongest associations (odds ratio = 3.98; $P = 0.003$). However, only seven (10%) of the 70 mammary GSOR hits located within GWAS-Marked windows contained the lipid metabolism GO term. Therefore, the power of annotation analyses like GO might be affected by imperfect knowledge of the multiple roles a gene plays, especially in species other than humans. These genes are potential candidate causal genes for PL phenotypes.

The mammary GSOR hits that were prioritized as candidate causal genes were *DGAT1*, *ABHD3*, *SERINC5*, *CYP2U1*, *PIGL*, *ARV1*, and *ACSM5*. Our findings revealed that the expression of *DGAT1* in the mammary gland is associated with different PL phenotypes, including phosphatidylserine, sphingolipids sphingomyelin, phospholipids phosphatidylcholine, phosphatidylinositol, and glucosylceramide phenotypes. The *DGAT1* gene has been reported to significantly influence the phenotypic variance of milk yield and composition in cattle [24,25]. The size of fat globules is a key determinant of polar lipids in milk; the smaller they are, the more membrane there are. This may help explain the influence of genes, such as *DGAT1*, that affect milk fatty acid composition. Although a protein-coding mutation for this gene has already been identified [25,26], the present study and previous ones [21,24,27] revealed *cis*-regulatory effects for *DGAT1*, possibly attributable to multiple causal mutations. GSOR analysis of mammary tissue found *ABHD3* on BTA24 to be significant; we observed the 2^nd most significant GWAS_{Meta} peak surrounding this gene. In cattle, a low-resolution GWAS found a broad region on BTA 24, including *ABHD3*, among many other genes associated with milk fatty acid concentration [28]. In a human GWAS, this gene was associated with circulating phospho- and sphingolipid concentrations [29], with a recent GWAS on plasma lipidome reporting a missense mutation causing this association [30]. In a metabolomics study on *ABHD3*, this gene was annotated as a lipase targeting medium-chain and oxidatively truncated phospholipids, establishing its physiological role in lipid metabolism [31]. This is the first time that a *cis*-regulatory mechanism linking the expression of the *ABHD3* gene in the mammary gland to the concentration of PLs in milk is reported. Our findings showed that the expression of the *ABHD3* gene in the mammary gland is associated mainly with sphingolipids sphingomyelin and phospholipids phosphatidylcholine but also associated with phosphatidylethanolamine and phosphatidylserine. Another gene is *ACSM5* (Acyl-CoA Synthetase Medium-Chain Family Member 5) on BTA 25, whose expression in the mammary gland is associated mainly with SM phenotypes but also with one of the phosphatidylethanolamine phenotypes. *ACSM5* catalyzes the activation of fatty acids by CoA to produce an acyl-CoA, which is the first step in fatty acid metabolism. *ACSM5* is involved in the fatty acid biosynthetic process and the acyl-CoA metabolic process (https://www.genecards.org/). The *ARV1* gene on BTA28, whose expression in the mammary gland is associated with glucosylceramide phenotype, is listed with several GO terms, including sphingolipid, cholesterol, bile acid metabolism, and cholesterol and sterol transport. It has been reported that Yeast cells lacking the *ARV1* gene harbor defects in sphingolipid metabolism [32].

Our study has some limitations. First, despite the relatively simple physiology and higher heritability of PL traits, the sample size used to predict GEBVs and estimate their correlations with gene expression was small. This could have limited the power of BayesR3 to identify variants with smaller effects on PL concentration. Although the sample size was small, our results demonstrate that we were still able to identify relevant genetic signals. However, we recommend testing the methods presented here with a larger dataset. Furthermore, a significant portion of the heritability of complex traits can be linked to *trans*-eQTLs, eQTLs located on different chromosomes or more than 5 Mb away [33], and these were not included in this study. However, a study on *trans*-eQTLs requires a large number of expression reference samples to ensure adequate statistical power.

In conclusion, the significant overlap between the genes identified by GWAS, GSOR, and GO indicated that all three methods have some power to identify genes mediating QTL. However, the odds ratios in Tables 3 and 4 are not very high, so the power of these methods is limited. A larger sample size could increase power, but we anticipate that some issues, such as LD, would persist. However, the combination of methods does give a list of candidate genes with fewer false positives.

## Materials and methods

### Ethics statement

No new animal experiments were undertaken for this study. All data were obtained from previously published studies.

### GWAS data description

Phenotypic data for Australian Holstein cows, including records for the concentration of 59 species of PLs in milk, were obtained. All experimental cows were maintained in the research herd at the Department of Economic Development, Jobs, Transport and Resources' Ellinbank Centre in Victoria, Australia, and the experimentation was conducted in accordance with the Australian Code of Practice for the Care and Use of Animals for Scientific Purposes. Cow diet varied through the milking season, but the majority of the cows' nutrient intake was usually derived from grazed pasture supplemented with bought-in feedstuff fed according to different strategies.

Three hundred sixty multiparous Holstein cows that calved in late winter/early spring were used in this study. The experiment was conducted over three years (2013, 2014, and 2015), with 120 cows participating each year. Milk samples were collected each year in three batches (40 animals per batch) over the period of mid-October to late-November. On each sampling occasion, the total milk from the afternoon and morning milking was collected into test buckets, pooled for each cow, and a subsample was taken for analysis. Milk samples were transported to the laboratory on ice and kept at -80 °C before analysis.

Polar lipids were extracted from raw milk as previously described [11]. Internal standard (PS 34:0) was added prior to lipid extraction. An Agilent 1290 UPLC system coupled to an LTQ-Orbitrap MS (Thermo Scientific) was used for polar lipid quantification. Chromatographic separation of polar lipids was achieved using a Luna HILIC column (250 × 4.6 mm, 5 µm, Phenomenex) maintained at 30 °C. The mobile phase was composed of 5 mM aqueous ammonium formate (A) and acetonitrile containing 0.1% formic acid (B). The flow rate was 0.6 mL/min with a gradient elution of 2–21% A over 25 min. The injection volume was 5 µL. The detection of lipids was by LTQ-Orbitrap mass spectrometer (Thermo Scientific) operated in electrospray ionization positive (for most polar lipid classes) or negative (for analysis of PI) Fourier transform mode. The resolution was set to 60,000 for both positive and negative modes. Identification of lipid species present in milk was performed as previously reported [11]. The quantification of selected polar lipid species was based on the peak area of parent ions after normalization based on the internal standard.

The GWAS data for milk polar lipid traits are publicly available at the Dryad data repository [34]. Heritability ($h^2$) was estimated for these traits using --reml command in GCTA [35]. The following model was used: $y = Xb + g + e$, where $y$ is a vector of phenotypic records; $X$ is a matrix of covariates (i.e., the combined effect of batch and year with nine levels); $g$ is a vector of random polygenic effects with $g \sim N(0, G\sigma_g^2)$, where $G$ is the genomic relationship matrix [36] estimated using 50K SNP genotypes, and $\sigma_g^2$ is the additive genetic variance explained by the 50K SNPs, and $e$ is a vector of residuals with $e \sim N(0, I\sigma_e^2)$, where $I$ is an identity matrix and $\sigma_e^2$ is residual variance. Heritability is defined as a ratio of the variance of genetic effects to the total variance.

Three PLs were excluded due to zero heritability, leaving 56 PLs for GWAS analysis (Table 1). Records with ±4 SD from the mean phenotype were excluded from the analysis (i.e., for 15 PL traits, the number of excluded records ranged from 1 to 6; for the remaining 41 PL, no records were excluded for this reason).

The 336 cows had already been genotyped because various subgroups had been involved in previous experiments in this government research herd: 181 were genotyped with Standard 50K SNP chips, 17 cows were genotyped with High Density (HD) 700K, the remainder with Low Density ~ 7.5K. Genotype data for the 336 cows was imputed to the whole genome sequences (WGS) level (Table 1). Minimac3 [37] was used to impute genotypes with Run7 of the 1000 Bull Genomes project as a reference population [38]. The details of the imputation are described in [39].

## Expression reference panel data

Two different sets of transcriptomics data were used from WBC and mammary tissues.

WBC gene expression was obtained from 143 Australian Holstein animals, a subset of a larger multibreed dataset. These cows were chosen from a single farm (the Ellinbank Smart Farm), and the only criterion was that they were lactating cows. However, DIM was fitted as a fixed effect. The processing of samples, RNA extraction, library preparation, RNA sequencing, etc., were described in detail in [33,40,41] (Table 1). This data did not overlap with the GWAS dataset. Imputation to WGS genotypes for the WBC RNA-seq animals was conducted using Run7 of the 1000 Bull Genomes project as a reference population [38] as described above for the GWAS animals.

Mammary gene expression data was obtained for 169 New Zealand Holstein cows [42–44](Table 1). For this dataset, 12,622,468 variants were imputed using 1,298 reference animals (including 306 Holstein-Friesian, 219 Jersey, 717 HF × J, and 56 other breeds) as described in [45].

## GWAS analysis

Variants with low imputation accuracy (Minimac $R^2 < 0.5$), minor allele frequency (MAF) < 0.01, and variants deviating from Hardy-Weinberg Equilibrium with $\chi^2$ $p$-value < 1E-6 were excluded. Single-trait GWAS were conducted for 56 PL phenotypes (S1 Data) using 14,056,074 imputed variants. For GWAS, we used a mixed linear model leaving-one-chromosome-out (MLM-LOCO) approach implemented in GCTA software [35,46]. GCTA initially adjusted the phenotypes for fixed effects (the combined effects of batch and year with nine levels) and then used the following model:

$$\mathbf{y} \;=\; \mathbf{1}\mu + \mathbf{bx} + \mathbf{g}^- + \mathbf{e} \tag{1}$$

where **Y** is a vector of records for adjusted phenotypic values; **1** is a vector of ones; μ is the mean of the trait; **b** is the additive allelic substitution effect of the SNP being tested; **x** is a vector of allele dosages (coded as 0, 1 or 2); $\mathbf{g}^-$ is a vector of polygenic effects with $\mathbf{g}^- \sim N(0, \mathbf{G}\sigma_g^2)$, where **G** is the genomic relationship matrix [36] calculated using SNP 50K genotypes, excluding variants located on the chromosome harboring the test SNP (LOCO approach); $\sigma_g^2$ is the additive genetic variance explained by the 50K SNPs; and **e** is the vector of residuals with $\mathbf{e} \sim N(0, \mathbf{I}\sigma_e^2)$, where **I** is an identity matrix and $\sigma_e^2$ is residual variance. To account for multiple testing issues, obtained $p$-values were adjusted using the Benjamini-Hochberg method [47] implemented in the *p.adjust* R function [48]. Variants with FDR ≤ 0.10 were treated as significant.

We also calculated the genomic inflation factor (λ). To do this, we transformed GWAS $p$-values to the chi-squared ($\chi^2$) test statistics using the quantile function distribution with one degree of freedom. Then, we used the formula $\lambda = \frac{media(\chi^2)}{0.455}$, where the numerator is the median of the observed chi-squared test statistics, and the denominator is the expected median of the chi-squared distribution under the null hypothesis.

## Multi-trait GWAS meta-analysis

We performed a meta-analysis using outputs from 56 single-trait GWAS (hereinafter referred to as GWAS$_{Meta}$) [18]. The following formula for GWAS$_{Meta}$ was used [18]:

$$\chi_i^2 = \mathbf{t_i}' \, \mathbf{V}^{-1} \, \mathbf{t_i} \tag{2}$$

where $\mathbf{t_i}$ is an n × 1 vector of signed t-values of $SNP_i$, and n is the number of traits used; $\mathbf{t_i'}$ is the transpose of $\mathbf{t_i}$ and $\mathbf{V^{-1}}$ is the inverse of a $n \times n$ correlation matrix where the correlation between two traits is the correlation over the 14,056,074 estimated SNP effects (signed t-values) of the two traits; $\chi_i^2$ is the chi-squared statistics with $n$ degrees of freedom for the i[th] SNP; the p-value for the i[th] SNP was calculated using the *pchisq* R function [48] with $n$ degrees of freedom [18]. To account for multiple testing issues, obtained p-values were adjusted using the Benjamini-Hochberg method [47] implemented in the *p.adjust* R function [48]. Variants with FDR ≤ 0.10 were treated as significant in GWAS$_{Meta}$.

## Predicting GEBVs for expression reference panel cows

We performed LD-pruning on genotypes of GWAS cows using PLINK v1.9 [49] with parameters --indep-pairwise 5000 500 0.95 to exclude variants that were in strong LD ($r^2 > 0.95$). The LD-pruned GWAS dataset with 1,236,780 SNP was used to train models with BayesR3 software [50] to predict GEBVs for the PL traits on the expression reference panel cows (Table 1). Individuals from the expression reference panel had the same set of 1,236,780 SNP genotypes. For each individual PL, the following model was used:

$$y = Xu + Vg + a + e \tag{3}$$

where $y$ is an $n \times 1$ column vector of phenotypic records, in which $n$ is the number of records; $X$ is an $n \times m$ incidence matrix, $u$ is $m \times 1$ vector of fixed effect and $m$ is the number of fixed effects including the combined effect of batch and year (nine levels); $V$ is the coded genotype, representing the observed genotypes of each individual; $g$ is a vector of SNP effects; $a$ is a vector of random genetic effects not explained by the SNPs with polygenic variance represented as $\sigma_a^2$, in which $a \sim N(0, A\sigma_a^2)$, and $A$ is the relationship matrix; and $e$ is the residual term. BayesR3 was run with 50,000 MCMC iterations and 25,000 burn-in. In the BayesR3 model, the SNP effects follow a mixture of four normal distributions with zero mean and additive genetic variances of zero, 0.0001, 0.001, and 0.01 times the genetic variance. Starting values for proportions of the four SNP effect distributions were defined as 0.994, 0.0055, 0.00049, and 0.00001, respectively.

Once SNP effects on an individual PL phenotype were estimated, local GEBVs for cows in the expression reference panel corresponding to a specific gene were calculated using the effect of SNPs positioned within ±1 Mb of that gene's TSS.

## Genetic Score Omics Regression (GSOR)

For an individual PL and RNA-seq dataset (i.e., tissue), the following per-gene GSOR model was applied:

$$\widehat{\mathbf{gebv_{local}}} = \mathbf{b_1}\,\Omega + \mathbf{b_2}\mathbf{X} + \mathbf{g} + \mathbf{e} \tag{4}$$

where $\widehat{\mathbf{gebv_{local}}}$ is an $m \times 1$ vector of local GEBVs (corresponding to a gene); $\Omega$ is a $m \times 1$ vector of that gene's expression; and m is the number of animals; $\mathbf{b_1}$ is the regression coefficient of the $\widehat{\mathbf{gebv_{local}}}$ on $\Omega$; $\mathbf{X}$ represents a design matrix for fixed effects (see next paragraph), and $\mathbf{b_2}$ is the vector of fixed effect (for the RNA-seq data); and $\mathbf{g}$ is a vector of random polygenic effects (for the RNA-seq data) with $\mathbf{g} \sim N(0, \mathbf{G}\sigma_g^2)$, where $\mathbf{G}$ is the genomic relationship matrix [36] calculated using 50K SNP genotypes, and $\sigma_g^2$ is the additive genetic variance explained by the 50K SNPs; $\mathbf{e}$ is the vector of residuals with $\mathbf{e} \sim N(0, \mathbf{I}\sigma_e^2)$, where $\mathbf{I}$ is an identity matrix, and $\sigma_e^2$ is residual variance.

Models for the WBC RNA-seq dataset were fitted using the experiment (with five levels corresponding to sampling time) as a categorical fixed effect and days in milk (DIM) as a quantitative fixed effect, with a mean and SD of 86 (± 36) days. No fixed effects were required to be fitted for the mammary RNA-seq dataset. Once associations between all the genes within an RNA-seq dataset and an individual PL were estimated, p-values were adjusted for multiple testing problem [47], conducted separately for each trait. Genes with FDR ≤ 0.01 were considered significant.

## Functional annotation analyses

Gene-set enrichment analyses were performed for each tissue separately using GSOR genes that were significant across 56 PL phenotypes, where the total number of genes (tested genes) within each RNA-seq dataset was used as background genes. We used the DAVID (The Database for Annotation, Visualization, and Integrated Discovery) bioinformatic tool [9,10] and regarded biological terms with FDR ≤ 0.05 as significant.

## Do GSOR hits agree with GWAS$_{Meta}$ signals?

We partitioned the genome into 100 Kb and 500 Kb non-overlapping windows, of which we identified those containing at least one significant GWAS$_{Meta}$ SNP (hereinafter referred to as GWAS-Marked windows). We then counted the GWAS-Marked windows and GSOR hits found within these windows. A GSOR hit is considered to be within a GWAS-Marked window if that gene's start position falls within that window. These values were compared to the total number of non-overlapping windows and the total number of GSOR hits. The Fisher Exact Test was used and a $P$-value ≤ 0.05 was considered significant.

## Are GSOR hits found within GWAS-Marked windows enriched for relevant GO terms?

We calculated the abundance of lipid genes (i.e., genes involved in Lipid metabolism GO) for two groups of genes, including (1) GSOR hits found within GWAS-Marked windows and (2) background genes (i.e., the total number of tested genes in that RNA-seq data). We labelled genes of each group as involved_in if the gene exists in the lipid metabolism GO and not_involved_in otherwise. We used the Fisher Exact Test to investigate the significance of the difference between the two groups. We regarded a $P$-value ≤ 0.05 as significant.

## Supporting information

**S1 Data. REML-based heritability estimations and the genomic inflation factor for GWAS for 56 milk polar lipid phenotypes.**
(CSV)

**S2 Data. Summary statistics from GWAS$_{Meta}$ for 56 PL phenotypes.**
(ZIP)

**S3 Data. GSOR $p$-values calculated using mammary dataset.** The first column is Gene Ensemble ID, and from the 2nd column onward are GSOR $p$-values (each column corresponds to one specific PL).
(TXT)

**S4 Data. The 1,404 GSOR hits identified for mammary transcriptome.**
(TXT)

**S5 Data. GSOR $p$-values calculated using white blood cells (WBC) dataset.** The first column is Gene Ensemble ID, and from the 2nd column onward are GSOR $p$-values (each column corresponds to one specific PL).
(TXT)

**S6 Data. The 2,186 GSOR hits identified for white blood cells transcriptome.**
(TXT)

## Acknowledgments

We thank Dr. Bolormaa Sunduimijid for imputation of sequence variants for cattle with Polar Lipid phenotypes and the partners from Run7 of the 1000 Bull Genomes Project for data access.

## Author contributions

**Conceptualization:** Michael E. Goddard.

**Data curation:** Iona M. Macleod, Amanda J. Chamberlain, Zhiqian Liu, Thomas J. Lopdell, Mathew D. Littlejohn.

**Formal analysis:** Mohammad Ghoreishifar.

**Methodology:** Mohammad Ghoreishifar, Michael E. Goddard.

**Project administration:** Jennie E. Pryce.

**Software:** Ruidong Xiang, Michael E. Goddard.

**Supervision:** Amanda J. Chamberlain, Jennie E. Pryce, Michael E. Goddard.

**Validation:** Mohammad Ghoreishifar.

**Writing – original draft:** Mohammad Ghoreishifar.

**Writing – review & editing:** Mohammad Ghoreishifar, Iona M. Macleod, Amanda J. Chamberlain, Zhiqian Liu, Ruidong Xiang, Jennie E. Pryce, Michael E. Goddard.

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
