## [Decision Letter · Decision Letter 0]

31 Dec 2024

PGENETICS-D-24-01281

An integrative approach to prioritize candidate causal genes for complex traits in cattle

PLOS Genetics

Dear Dr. Ghoreishifar,

Thank you for submitting your manuscript to PLOS Genetics. After careful consideration, we feel that it has merit but does not fully meet PLOS Genetics's publication criteria as it currently stands. Therefore, we invite you to submit a revised version of the manuscript that addresses the points raised during the review process.

Please submit your revised manuscript within 60 days Mar 01 2025 11:59PM. If you will need more time than this to complete your revisions, please reply to this message or contact the journal office at plosgenetics@plos.org. Please include the following items when submitting your revised manuscript:

We look forward to receiving your revised manuscript.

Kind regards,

Martien Groenen, PhD

Academic Editor

PLOS Genetics

Giorgio Sirugo

Section Editor

PLOS Genetics

Aimée Dudley

Editor-in-Chief

PLOS Genetics

Anne Goriely

Editor-in-Chief

PLOS Genetics

**Additional Editor Comments:**

Two reviewers comment on the limited power and generality of the GWAS performed. They also, comment on the fact the manuscript describing the method has been deposited in BioRxiv but apparently still not been published. In your revised manuscript you should pay particular attention addressing these aspects.

**Journal Requirements:**

At this stage, the following Authors/Authors require contributions: Mohammad Ghoreishifar, Iona M. Macleod, Amanda J. Chamberlain, Zhiqian Liu, Thomas J. Lopdell, Mathew D. Littlejohn, Ruidong Xiang, Jennie E. Pryce, and Michael E. Goddard. Please ensure that the full contributions of each author are acknowledged in the "Add/Edit/Remove Authors" section of our submission form.

The list of CRediT author contributions may be found here: https://journals.plos.org/plosgenetics/s/authorship#loc-author-contributions

https://journals.plos.org/plosgenetics/s/submission-guidelines#loc-parts-of-a-submission

3) We noticed that you used the phrase 'data not shown' in the manuscript. We do not allow these references, as the PLOS data access policy requires that all data be either published with the manuscript or made available in a publicly accessible database. Please amend the supplementary material to include the referenced data or remove the references.

5) We note that your Data Availability Statement is currently as follows: "All gene expression data was taken from previously published studies as detailed in the materials and methods. The Polar Lipids data used in the study though taken from a previously published study (see materials and methods) had not been publicly released. This data has been uploaded to the Dryad repository and will become publicly accessible upon acceptance of this paper.". Please confirm at this time whether or not your submission contains all raw data required to replicate the results of your study. Authors must share the “minimal data set” for their submission. PLOS defines the minimal data set to consist of the data required to replicate all study findings reported in the article, as well as related metadata and methods (https://journals.plos.org/plosone/s/data-availability#loc-minimal-data-set-definition).

- The points extracted from images for analysis..

6) Please ensure that the funders and grant numbers match between the Financial Disclosure field and the Funding Information tab in your submission form. Note that the funders must be provided in the same order in both places as well. State the initials, alongside each funding source, of each author to receive each grant. For example: "This work was supported by the National Institutes of Health (####### to AM; ###### to CJ) and the National Science Foundation (###### to AM).".

**Reviewers' comments:**

Reviewer's Responses to Questions

**Comments to the Authors:**

Reviewer #1: See attachment

Reviewer #2: This is a very interesting, clear and well written article. The main originality is the use of the GSOR method. However, this method has already been described in a paper (by the same team) deposited on BioRxiv, and therefore presumably submitted for publication.

If I understand correctly, the present article is therefore an application of this method. For a good demonstration, the GWAS dataset does not seem to me to be the most appropriate. Of course, the traits are selected for their assumed simple determinism, which limits the number of expected QTL. But the limited size of the dataset does not prove the universal effectiveness of the method: GWAS have limited power and the partial genomic values are likely to have very low accuracy. For a more convincing demonstration, I would have preferred to see a larger application.

Note: An important determinant of polar lipids in milk is the size of the fat globules. The smaller they are, the more membranes there are. I think this point could be discussed. And fat globule size is much easier to measure than polar lipids. This could also explain the influence of genes that affect fatty acids, such as DGAT1.

Minor remarks

23: Do you mean : A multi-trait meta-analysis of GWAS… ?

117: standard errors of h2 estimates seem quite optimistic with regards to the number of records

123: because Methods are presented after Results, it would be useful to define GWASMeta

367: I do not understand g in equation (4). Why explaining partial genomic values by whole genomic values?

Reviewer #3: Ghoreishifar et al. conducted a GWAS for 56 traits related to polar lipids in milk, utilizing three main sources of information: GWAS (including meta-GWAS hits), genetic score omics regression (GSOR), and GO analysis. Although the sample size for the GWAS is relatively small, some of the studied phenotypes are expected to exhibit simple genetic architectures, making it possible for the study to identify meaningful associations.

A major concern, however, is that the GWAS, meta-GWAS, and GSOR analyses are all derived from the same dataset, which means these sources are not independent, potentially limiting the robustness of the findings.

Regarding the GSOR methodology, the manuscript (overlapping authors with the current manuscript) has been available on bioRxiv since July 15, 2022. Wondering why it has not yet been published in a peer-reviewed journal.

Line 62: It is worth noting that the association is not only on the same chromosome but also that the variant and the gene are closely located.

Line 93. “these traits are simpler” -> these traits are expected to have simpler genetic architecture

Line 121: How was this P-value threshold reached?

Line 123-123. Did you examine how many independent QTL are segregating on these 2 chromosomes?

Line 129-130: did you run multi-trait meta-analysis for 21 PLs having ‘significant association’ in single-trait analysis?

Line 232-233: “Seven of the 70 mammary 233 GSOR hits located within GWAS-Marked windows contained the lipid metabolism GO term” – could you comment what are the other possible physiological mechanisms linking to PLs for the rest of GSOR hits (90%).

Line 239-241. “The DGAT1 gene has been reported to account for 30-40% of the phenotypic variance of milk yield and composition in cattle [23, 24]”. Please check it carefully; r^2 (QTL) is 0.18 for milk yield (Grisart et al. 2002).

L317-330: The genomic inflation factors (lambda values) for GWAS should be presented (maybe as supplemental table), as LOCO approach can increase genomic inflation (https://doi/10.1111/jbg.12419).

L331-342: Multi-trait meta-analysis: There are two issues. The LD score regression approach (https://doi.org/10.1038/ng.3406) should be used for estimation of genetic correlation to avoid bias due to sample overlap and spread of LD. For similar reasons, i.e. the GWAS was done in one population for all the traits, approach like MTAG (https://doi.org/10.1038/s41588-019-0469-9) . should be employed.

**Have all data underlying the figures and results presented in the manuscript been provided?**

Reviewer #1: **No: **

Reviewer #2: **No: ** The authors state that the polar lipids phenotypes will be released upon acceptance.

Reviewer #3: **No: ** It says "The Polar Lipids data used in the study though taken from a previously published study (see materials and methods) had not been publicly released. It will be publicly available upon acceptance of this paper."

PLOS authors have the option to publish the peer review history of their article (what does this mean? ). If published, this will include your full peer review and any attached files.

**Do you want your identity to be public for this peer review?** For information about this choice, including consent withdrawal, please see our Privacy Policy .

Reviewer #1: No

Reviewer #2: No

Reviewer #3: No

**Figure resubmission:**
---

## [Decision Letter · Decision Letter 1]

23 Mar 2025

PGENETICS-D-24-01281R1

An integrative approach to prioritize candidate causal genes for complex traits in cattle

PLOS Genetics

Dear Dr. Ghoreishifar,

Thank you for submitting your manuscript to PLOS Genetics. After careful consideration, we feel that it has merit but does not fully meet PLOS Genetics's publication criteria as it currently stands. Therefore, we invite you to submit a revised version of the manuscript that addresses the points raised during the review process.

Please submit your revised manuscript within 60 days May 22 2025 11:59PM. If you will need more time than this to complete your revisions, please reply to this message or contact the journal office at plosgenetics@plos.org. Please include the following items when submitting your revised manuscript:

We look forward to receiving your revised manuscript.

Kind regards,

Martien Groenen, PhD

Academic Editor

PLOS Genetics

Giorgio Sirugo

Section Editor

PLOS Genetics

Aimée Dudley

Editor-in-Chief

PLOS Genetics

Anne Goriely

Editor-in-Chief

PLOS Genetics

**Additional Editor Comments:**

While two of the reviewers think that in the revised manuscript their comments were addressed satisfactorily, reviewer 3 points to a number of potential difficulties that have not been addressed sufficiently. In particular the possibility that many of the identified hits might be potential false positives needs to be better addressed as this is central to the claim that the overlap between GWAS, GSOR, and GO analyses suggest would provide increased power to identify genes mediating QTL. Given the small sample size of the study (which is still commented to be a weakness of the study by two of the reviewers) the specific comments made by reviewer 3 need to be better addressed before the manuscript can be considered for publication.

**Journal Requirements:**

1) We noticed that you used the phrase 'data not shown' in the manuscript. We do not allow these references, as the PLOS data access policy requires that all data be either published with the manuscript or made available in a publicly accessible database. Please amend the supplementary material to include the referenced data or remove the references.

2) In the online submission form, you indicated that your data will be submitted to the Dryad database upon acceptance. Should your submission be accepted, we will require the following information in your Data Availability Statement:

1. The DOI provided by Dryad

2. The citation for your data package in the reference section of your manuscript

3. The citation for your data package in the methods section

If you are unable to adhere to our open data policy, please kindly revise your statement to explain your reasoning and we will seek the editor's input on an exemption. Please be assured that, once you have provided your new statement, the assessment of your exemption will not hold up the peer review process.

3) Please amend your detailed Financial Disclosure statement. This is published with the article. It must therefore be completed in full sentences and contain the exact wording you wish to be published.

4) Please ensure that the funders and grant numbers match between the Financial Disclosure field and the Funding Information tab in your submission form. Note that the funders must be provided in the same order in both places as well. Currently, the order of the funders is different in both places.

**Reviewers' comments:**

Reviewer's Responses to Questions

Reviewer #1: Thank you for addressing my comments

Reviewer #2: The authors responded satisfactorily to my comments.

Reviewer #3: The authors have revised the manuscript but there are several shortcomings which were not addressed satisfactorily.

“Please note that all RED highlights in the manuscript are modifications made to improve written language, and the BLUE ones are modifications made to address the reviewers’ comments”

I don’t see any red or blue highlighted text in the revised manuscript (pdf version). Or is it only me?

The major shortcomings are that the GWAS population is small and large number of traits (56) were analyzed. The manuscript describing the GSOR method is yet to be published as a peer-reviewed article. The results observed cannot be generalized as “comparatively simple series of traits” were selected.

“As mentioned in the paper (lines 143 and 146-149), we run multi-trait meta-analyses using all the 56 PL phenotypes.”

Is it reasonable to assume the same genetic variant is affecting all 56 traits?

“Line 232-233: “Seven of the 70 mammary GSOR hits located within GWAS-Marked windows contained the lipid metabolism GO term” – could you comment what are the other possible physiological mechanisms linking to PLs for the rest of GSOR hits (90%).

R: We found several GO terms significantly enriched by GSOR gene lists (see Table 2). However, they were not directly related to PL, and the overlapping analyses between GWAS and GSOR could not find any significant term other than polar lipids. They could potentially be false positives.”

If that is the case it seems too many false positives!

“R: The LD score regression doesn’t work in cattle because of the extensive LD. We tried them and it was useless.

R: The multi-trait meta-analyses method we used has been published and been used many times and we don’t see that there is a problem with it.”

It is a very strange response. The second statement implies that a published article cannot be questioned. Contrary “We tried them and it was useless” is very vague without substantiating.

**Have all data underlying the figures and results presented in the manuscript been provided?**

Reviewer #1: Yes

Reviewer #2: None

Reviewer #3: Yes

PLOS authors have the option to publish the peer review history of their article (what does this mean? ). If published, this will include your full peer review and any attached files.

**Do you want your identity to be public for this peer review?** For information about this choice, including consent withdrawal, please see our Privacy Policy .

Reviewer #1: No

Reviewer #2: No

Reviewer #3: No

**Figure resubmission:**
---

## [Editor Report · Decision Letter 2]

8 May 2025

Dear Dr Ghoreishifar,

We are pleased to inform you that your manuscript entitled "An integrative approach to prioritize candidate causal genes for complex traits in cattle" has been editorially accepted for publication in PLOS Genetics. Congratulations!

Yours sincerely,

Martien Groenen, PhD

Academic Editor

PLOS Genetics

Giorgio Sirugo

Section Editor

PLOS Genetics

Aimée Dudley

Editor-in-Chief

PLOS Genetics

Anne Goriely

Editor-in-Chief

PLOS Genetics

Comments from the reviewers (if applicable):

The remaining comments of reviewer 3 have been sufficiently addressed.

**Data Deposition**

http://datadryad.org/submit?journalID=pgenetics&manu=PGENETICS-D-24-01281R2

**Press Queries**

---

## [Editor Report · Acceptance letter]

PGENETICS-D-24-01281R2

An integrative approach to prioritize candidate causal genes for complex traits in cattle

Dear Dr Ghoreishifar,

We are pleased to inform you that your manuscript entitled "An integrative approach to prioritize candidate causal genes for complex traits in cattle" has been formally accepted for publication in PLOS Genetics! Your manuscript is now with our production department and you will be notified of the publication date in due course.

With kind regards,

Zsofia Freund

PLOS Genetics

On behalf of:
